# Characterization of Microtubule Destabilizing Drugs: A Quantitative Cell-Based Assay That Bridges the Gap between Tubulin Based- and Cytotoxicity Assays

**DOI:** 10.3390/cancers13205226

**Published:** 2021-10-18

**Authors:** Marie-Catherine Laisne, Sophie Michallet, Laurence Lafanechère

**Affiliations:** Institute for Advanced Biosciences, Team Cytoskeletal Dynamics and Nuclear Functions, INSERM U1209, CNRS UMR5309, Université Grenoble Alpes, 38000 Grenoble, France; marie-catherine.laisne@univ-grenoble-alpes.fr (M.-C.L.); sophie.michallet@univ-grenoble-alpes.fr (S.M.)

**Keywords:** cancer therapy, microtubules, cell-based assay, drug discovery

## Abstract

**Simple Summary:**

The characterization of new microtubule depolymerizing agents relies mainly on purified tubulin assays in vitro and on cytotoxicity tests. However, the relationship between the in vitro effects of drugs and their effect on cell viability may not be direct. Here, we have systematically compared the effect of four reference drugs on tubulin polymerization in vitro and in cells, using a recently-developed quantitative assay of the cellular microtubule content. By comparing these results with cell viability assays, we found that this new cellular microtubule content test better predicts cellular drug toxicity than the in vitro tubulin polymerization assay. This test can thus be easily implemented in the process of discovery and characterization of novel microtubule poisons.

**Abstract:**

(1) Background: Microtubule depolymerizing agents (MDAs) are commonly used for cancer treatment. However, the therapeutic use of such microtubule inhibitors is limited by their toxicity and the emergence of resistance. Thus, there is still a sustained effort to develop new MDAs. During the characterization of such agents, mainly through in vitro analyses using purified tubulin and cytotoxicity assays, quantitative comparisons are mandatory. The relationship between the effect of the drugs on purified tubulin and on cell viability are not always direct. (2) Methods: We have recently developed a cell-based assay that quantifies the cellular microtubule content. In this study, we have conducted a systematic comparative analysis of the effect of four well-characterized MDAs on the kinetics of in vitro tubulin assembly, on the cellular microtubule content (using our recently developed assay) and on cell viability. (3) Conclusions: These assays gave complementary results. Additionally, we found that the drugs’ effect on in vitro tubulin polymerization is not completely predictive of their relative cytotoxicity. Their effect on the cellular microtubule content, however, is closely related to their effect on cell viability. In conclusion, the assay we have recently developed can bridge the gap between in vitro tubulin assays and cell viability assays.

## 1. Introduction

Microtubules (MTs) are dynamic structures that are involved in intracellular trafficking, cell shape, cell movements and the segregation of condensed chromosomes during mitosis. They are composed of α-β tubulin heterodimers and their polymerization exhibits non-equilibrium dynamics, characterized by periods of polymerization and of depolymerization. This intrinsic MT dynamics is tightly controlled in the cell by interaction with an array of proteins, such as XMAP215/Dis1/TOGp, MCAK, MAP4, end-binding proteins or Op18/stathmine [1,2,3]. Targeted perturbation of this finely tuned process constitutes a major therapeutic strategy. Drugs that interfere with tubulin and MTs are, indeed, key components of combination chemotherapies for the treatment of carcinomas [4,5,6]. In addition to cancer, targeting the microtubular cytoskeleton can be an effective therapeutic strategy for a wide variety of diseases including neurodegenerative and mental diseases, viral infections and parasitic diseases [7,8,9,10,11,12].

Numerous compounds bind to tubulin and MTs. They can be roughly classified into MT stabilizing agents (MSAs) such as taxanes, and MT destabilizing agents (MDAs). Regarding MDAs, different binding sites located on the β-tubulin subunit have been identified for colchicine, vinca-alkaloids and maytansine. Additionally, the pironetine site has been found on the α-tubulin subunit [13].

The colchicine site is located at the intradimer interface [14,15,16]. It is a large site, mostly buried in the β-tubulin subunit. Besides colchicine, numerous drugs bind to this site, such as nocodazole or combretastatin, but no ligand large enough to occupy the entire site has yet been described [6]. Colchicine-site ligands destabilize MTs by preventing the curved-to-straight conformational transition within the αβ-tubulin heterodimer [13,17].

The vinca site is located at the inter-dimer interface between two longitudinally aligned tubulin dimers. Vinca-site MDAs destabilize MTs by introducing a wedge at the interface between two longitudinally aligned tubulin dimers at the tip of MTs, or by stabilizing assembly-incompetent ring-like oligomers of tubulin [13,18].

The maytansine site is distinct from the vinca and the colchicine sites. Binding of maytansin, spongistatin, PM60184, disorazole or rhizoxin to this site blocks the formation of longitudinal tubulin interactions in MTs [13,19].

Microtubule targeting agents (MTAs) have shown massive success in the clinic. However, their therapeutic use is hampered by their toxicity and the development of resistance [5]. There is thus still a sustained effort to develop new microtubule depolymerizing agents to address these issues. For instance, eribulin, a synthetic analog of the natural compound halichondrin B that binds at or near the vinca site [20], has been shown to induce less peripheral neuropathy than vinca alkaloids and has been approved for the treatment of metastatic breast cancers and liposarcomas [21,22]. Additionally, agents that target the colchicine site may hold promise for their anti-angiogenic properties [6,23]. 

The process of characterizing new MTAs is commonly based on the in vitro analysis of their binding to tubulin (binding assays, competition assays with known agents, affinity measurements [24,25,26,27,28]) as well as on the analysis of their effect on the polymerization of tubulin into microtubules (spectrometric monitoring of the kinetic of tubulin assembly [29,30] or, more recently, tracking of the growth of individual microtubules reconstituted in vitro, by time-lapse fluorescence microscopy [31]). The compounds’ characterization is then complemented using tests on cells, such as immunofluorescence analysis of the effect of the compounds on the microtubule network, FACS analysis of their ability to interfere with the cell cycle, and finally analysis of their cytotoxicity. 

Quantitative analyses are mandatory for the comparisons between different classes of agents as well as for the study of the structure activity relationship between structural analogs. Quantitative evaluation of the compounds’ effects is achievable using in vitro biochemical assays, in which the concentration of compounds and tubulin is controlled, as well as all other biochemical parameters (i.e., buffers, concentration of GTP, pH, temperature, etc.) [32].

Further quantitative comparison of the effect of compounds on cells are currently conducted by analyzing their antiproliferative activity, comparing the GI50, i.e., the compound concentration that inhibits 50% of cell growth, using cell viability assays. Cell viability assays allow to determine the number of living cells in a sample. The most common reporters of cell viability are vital dyes such as propidium iodide. Cell viability assays also typically measure the metabolic activity or ATP content of cells. 

These are global tests, the results of which are analyzed after a long time (several days) of incubation of the cells with the compounds. They have the advantage of providing information on the ability of a compound to penetrate into the cells and to act in a cellular context. However, cell death may result not only from the interaction of compounds with MTs, and the subsequent cascade of events leading to apoptosis, but also potentially from the interaction of compounds with other cellular targets. For example, several drugs, such as nocodazole or rigosertib, are high affinity ligands not only of tubulin, but also of kinases, which could account for their effect on cell viability [31,33]. Thus, the relationship between the efficiency of compounds assessed in tests on purified tubulin and their cytotoxicity is indirect. Assuming that cell death is the result of the sole effect of compounds on microtubule dynamics may thus be erroneous.

We have recently developed an immunoluminescent assay to measure the amount of MTs present in cells after a treatment with MDAs [29] or MSAs [34]. This quantitative test is easy to implement, reproducible and does not require a microscope but a simple microplate reader.

Here we have conducted a systematic comparative analysis of the effect of four well-characterized MDAs on the kinetics of in vitro tubulin assembly, on the cellular MT content (using our assay) and on cell viability.

The structures of the 4 reference compounds, colchicine, nocodazole, vinblastine and combretastatin-A4 (C-A4), are shown in Figure 1. Vinblastine binds to the vinca site of tubulin, whereas colchicine, nocodazole and C-A4 bind to the colchicine site. We have chosen three different compounds that bind to the colchicine site, because they differ in their kinetics of binding and unbinding: while colchicine binds and detaches slowly from tubulin, nocodazole and C-A4 have rapid binding kinetics and their binding is rapidly reversible [35,36]. Moreover, despite their ability to depolymerize microtubules, these compounds differ in their applications: nocodazole has no clinical application but remains an excellent research tool [36], colchicine, at low doses, is used in the treatment of inflammatory disorders [37], C-A4 is an anti-neoangiogenesis agent [38] while vinblastine is used in antitumor chemotherapy [39].

## 2. Materials and Methods

### 2.1. Chemical Reagents and Cells

Human HeLa cells, a cell line derived from cervical adenocarcima, and RPE-1 cells, which are human retinal pigment epithelial cells, were obtained from the American Type Culture Collection (ATCC, Gainthersburg, MD, USA). HeLa cells were grown in RPMI 1640 medium with Glutamax (Gibco Invitrogen, Carlsbad, CA, USA) and RPE-1 cells were grown in DMEM, 4.5 g/L glucose (Gibco Invitrogen, Carlsbad, CA, USA). Both media were supplemented with 10% fetal bovine serum and 1% penicillin-streptomycin. Cells were maintained in a humid incubator at 37 °C in 5% CO_2_.

All chemicals, except those for which it is specified, were purchased from Sigma-Aldrich (Saint-Quentin-Fallavier, France).

Nocodazole, combretastatin-A4, colchicine and vinblastine were prepared at a 10 mM stock solution in Dimethyl sulfoxide (DMSO, #4540) aliquoted and stored at −20 °C.

### 2.2. In Vitro Tubulin Polymerization Assay

Tubulin polymerization kit ((#BK004) was purchased from Cytoskeleton, Inc. (Denver, CO, USA). To study the effect of the compounds on MT assembly, the absorbance-based in vitro tubulin polymerization assay was performed following the manufacturer’s protocol. The reaction mixture containing porcine tubulin (40 µM) in G-PEM buffer (80 mM PIPES pH 6.9, 2.0 mM MgCl2, 0.5 mM EGTA, and 5% glycerol) supplemented with 1.0 mM GTP, in the presence of DMSO (0.18%) or the compound to be tested, was prepared and added to each well of a 96-well plate (half area, #675096, Greiner Bio One, Courtaboeuf, France). Tubulin polymerization was monitored by the increase in absorbance over time, measured by a CLARIOstar^plus^ Microplate reader from BMG Lab technology (Champagny-sur-Marne, France). The IC50 value of each compound was estimated graphically as the concentration which decreased the maximum assembly rate of tubulin by 50% compared to the rate in the presence of DMSO (100%).

### 2.3. Quantitative Assay of the Cellular Microtubule Content

7500 HeLa cells were seeded in 96-well microplates (#655086, Greiner bio One, Courtaboeuf, France) in 100 µL of complete medium per well and then incubated at 37 °C in 5% CO_2_ for 24 h. Cells were then treated for 30 min at 37 °C with the compounds at concentrations ranging from 1 to 5000 nM (1 microplate per molecule, 1 concentration per column), with 0.1% DMSO used as positive control (6 wells per microplate). After medium aspiration, treated cells were permeabilized for 10 min using 100 µL per well of warmed (37 °C) OPT buffer (80 mM Pipes, 1 mM EGTA, 1 mM MgCl_2_, 0.5% Triton X-100, and 10% glycerol, pH 6.8). Cells were fixed overnight at room temperature using 100 µL per well of 4% formaldehyde (Sigma Aldrich, #252549, Saint-Quentin-Fallavier, France) in PBS. Cells were washed 3 times in PBS, 0.1% Tween-20 (100 µL per well), then primary anti-alpha-tubulin antibody (clone α3A1 [40], 1:5000 in PBS 2% Bovine Serum Albumin (BSA)) was added for 45 min. Cells were washed twice again and secondary anti-mouse antibody coupled to HRP (1:2000 in PBS 2% BSA, #715-035-150, Jackson Immuno-Research Laboratories, Cambridgeshire, UK) was added for 45 min. Then, cells were washed again with PBS and 100 µL of ECL substrate (#170-5061, Bio-Rad Laboratories inc., USA) were injected in each well using the FLUOstar OPTIMA Microplate Reader (BMG Lab technology, Champagny-sur-Marne, France). The luminescent signal was read immediately after ECL injection. IC50s, i.e., drug concentrations able to reduce the amount of cellular microtubules by half, were calculated for each independent experiment using GraphPad Prism software and are presented in the text as means ± SEM.

### 2.4. Immunofluorescence

Cells at a density of 20,000 cells per well were grown for 48 h on glass coverslips placed in a 24-well microplate. When cells reached 70% confluence, the medium was replaced with a fresh one supplemented with DMSO (0.005% or 0.01%) or the test compound at 50 nM or 1000 nM. After 30 min of incubation, cells were permeabilized in warm OPT buffer (80 mmol/L Pipes, 1 mol/L EGTA, 1 mol/L MgCl_2_, 0.5% Triton X-100 and 10% Glycerol, pH 6.8) and fixed for 6 min in −20 °C methanol (Carlo ERBA SAS, #414855, Val-de-Reuil, France). After washing and saturation with a specific blocking buffer (3% BSA), 10% Goat serum (Gibco Invitrogen, #16210064, Carlsbad, CA, USA) in PBS), cells were incubated for 45 min at room temperature (RT) with anti-alpha-tubulin antibody (clone α3A1 [40] in blocking buffer). Cells were washed twice again and subsequently incubated with Alexa 488 conjugated anti-mouse antibody (1:500 in blocking buffer, #115-545-166, Jackson immune-research laboratory, Cambridgeshire, UK) and DNA was stained with Hoechst 33342 (1:10,000 in blocking buffer) for 30 min at RT. Coverslips were mounted on glass slides with Moviol 4–88.

Images were captured with a Zeiss AxioimagerM2 microscope equipped with the acquisition software AxioVision (Marly-le-Roi, France).

### 2.5. Analysis of Cell Viability

Cell viability was analyzed using the colorimetric PrestoBlue assay (Gibco Invitrogen, #A13262, Carlsbad, CA, USA). Cells were seeded in 96-well microplates (#655077, Greiner, Courtaboeuf, France) at density of 2500 cells per well and allowed to adhere for 24 h before being treated for 48 h with the drugs at indicated concentrations or corresponding concentrations of DMSO as controls, in a final volume of 90 µL. After 48 h of treatment, 10 µL PrestoBlue was added to each well and cells were incubated for another 45 min. The absorbance of each well was measured using a FLUOstar microplate reader (Excitation: 544 nm; Emission: 580 nm, BMG Lab technology, Champagny/Marne, France). GI50s, i.e., drug concentrations able to reduce cell growth by half, were calculated for each independent experiment using GraphPad Prism software and are presented in the text as means ± SEM.

## 3. Results

### 3.1. Effect of the Compounds on In Vitro Kinetics of Tubulin Polymerization

First, we compared the effect of different drugs on in vitro tubulin polymerization. The kinetics of MT assembly was followed by measuring the time course of the absorbance at 350 nm. As expected, all drugs inhibited tubulin polymerization in a dose-dependent manner (Figure 2).

However, while the same concentration of tubulin (40 µM) is used, differences in activity are observed between drugs. Thus, colchicine and vinblastine are the most potent drugs, with an IC50 close to 1 µM. C-A4 also shows a strong depolymerizing activity with an IC50 close to 2.5 µM. An almost complete depolymerization was achieved with about 10 µM of these 3 drugs. These results indicate that targeting the vinblastine or the colchicine site can give similar results on tubulin assembly. The depolymerizing action of nocodazole is less strong, with an apparent IC50 close to 5 µM. Moreover, an almost complete inhibition of tubulin is observed only for nocodazole concentrations higher than or equal to 40 µM. Thus, contrary to colchicine and C-A4, equimolar concentrations of nocodazole and tubulin are necessary to completely inhibit tubulin assembly. This suggests that, although nocodazole binds to the colchicine-binding site, its in vitro depolymerizing mechanism of action is different than that of colchicine. Indeed, the fact that sub-stoichiometric colchicine concentrations can induce a complete inhibition of tubulin assembly is a consequence of the kinetic blockage of the microtubule ends through the fixation of the slowly reversible tubulin-colchicine complex [41,42]. On the contrary, it takes as many nocodazole molecules as tubulin molecules to completely prevent the assembly of tubulin into microtubules, indicating that nocodazole acts mainly by inactivating free tubulin dimers [43].

All these results show that under strictly comparable assembly conditions, and while the concentration of MDAs is controlled, MDAs can have different consequences on the assembly kinetics.

### 3.2. Effect of the Compounds on Cellular Microtubules

We then compared the effect of the 4 compounds on cellular interphase MTs. We used our recently developed cell-based assay that quantifies intact MTs in HeLa cells using immunoluminescence [29]. The main steps of the assay are illustrated in Figure 3A and the detailed protocol is given in the methods section.

As shown on Figure 3B, all compounds induced a dose-dependent depolymerization of cellular MTs. Interestingly, these compounds can be grouped 2 by 2 with colchicine and nocodazole having respectively an IC50 of 786.67 ± 81.72 nM and 350.00 ± 76.38 nM, and C-A4 and vinblastine having much lower IC50 values, i.e., 4.50 ± 0.76 nM and 4.83 ± 0.17 nM, respectively. These differences are not the result of compounds binding to different tubulin sites, since C-A4 binds to the colchicine site, as does nocodazole and colchicine.

We checked using immunofluorescence that the luminescent values do reflect the state of cell MTs. As shown on Figure 4, a MT network is still present in cells treated with 50 nM of colchicine and nocodazole, whereas only few MTs remain when cells are treated with 50 nM vinblastine or C-A4. A treatment with a concentration of 1000 nM induced the depolymerization of almost all the MT network, whatever the compound used. Colchicine was found to be the less potent drug, reflecting the results measured with the microplate luminescence reader.

As HeLa cells are tumoral cells, we wondered if the lower sensibility of the cells to colchicine and nocodazole, as compared to C-A4 and vinblastine, could result from a different activity of the efflux pumps, involved in the resistance phenomena, toward these different drugs. To test this hypothesis, we analyzed by immunofluorescence the effect of these 4 drugs on a non-cancerous epithelial line, the RPE1 cells. As illustrated in Figure 5, the effects of the drugs are similar to those observed on HeLa cells: at a 50 nM concentration, only vinblastine and C-A4 depolymerized cell microtubules. This result thus suggests that the observed differences between the different drugs can be generalized to tumoral and non-tumoral epithelial cell lines.

### 3.3. Effects of the Compounds on Cell Viability

We further evaluated the effect of the compounds on HeLa (Figure 6A) and RPE-1 (Figure 6B) cells viability. To that aim we used the sensitive PrestoBlue^®^ fluorescent assay, which measures the number of metabolically active cells. This assay was performed after 48 h of incubation of the cells with different doses of the compounds. As shown in Figure 6, all compounds affected cell viability, in a dose-dependent manner.

Although the GI50 (i.e., the concentration causing 50% cell growth inhibition) values are approximately 5- to 20-fold lower than those observed in the quantitative assay measuring the effect of drugs on cellular MTs, the profile of the curves is strikingly similar to that observed in Figure 3B. For instance, regarding HeLa cells, vinblastine and C-A4 have a similar high cytotoxic activity with a GI50 of 0.73 ± 0.02 nM and 0.93 ± 0.07 nM, respectively, whereas those of colchicine and nocodazole are 10 to 50 times higher, with a GI50 of 9.17 ± 0.60 nM and 49.33 ± 2.60 nM, respectively. The GI50s observed for these different drugs on RPE-1 cells vary in a similar way, with a GI50 of 0.70 ± 0.77 and 4.16 ± 1.42 for vinblastine and CA-4 respectively, whereas those of colchicine and nocodazole have GI50 values of 30.00 ± 1.73 nM and 81.67 ± 4.41 nM, respectively.

## 4. Discussion

The development of agents targeting MTs remains an area of intense research. In this context, quantitative comparisons between new compounds and known reference agents, as well as between structural analogs for a structure–activity relationship study are essential.

The usual in vitro assay for the characterization of MDAs is an assembly assay where the polymerization of tubulin into MTs upon addition of GTP and temperature increase is followed using a spectrophotometer [32] or a fluorometer [44]. The extent of microtubule assembly is assessed as a function of ligand concentration, and the potency of the substance in question is calculated from these data. The amount of ligand required to observe an effect on tubulin assembly also depends on the concentration of tubulin and the type of buffer. Indeed, some buffers such as those based on PIPES favor tubulin assembly [45,46,47]. It is therefore important to compare the new compounds to standard agents under identical conditions of tubulin concentration and buffer.

Here, we have compared the in vitro effect on tubulin assembly of a set of known and well-characterized MDAs, under the same conditions. We have observed differences in the potency of the three different MDAs that have been described to bind to the colchicine-binding site; although colchicine and C-A4 show a similar potency, nocodazole was found less potent in this assay. Indeed, a concentration of nocodazole equimolar to the tubulin concentration was found necessary to completely inhibit tubulin assembly, indicating that nocodazole impacts tubulin polymerization by decreasing the concentration of assembly-competent tubulin dimers. On the contrary only sub-stoichiometric concentrations of colchicine and C-A4 are necessary to completely inhibit tubulin assembly. This is most probably due to the well-described kinetic blockage of the microtubule plus end by colchicine- or C-A4-tubulin complexes [41,43], hampering further polymerization. Although nocodazole binds at the same site as colchicine, a recent structural study showed that it does not occupy the pocket in the same way. Thus, nocodazole overlapped very little with colchicine. Unlike colchicine, nocodazole is located deeply in the tubulin β subunit and makes no interaction with the α subunit [48]. Such structural differences may have some consequences in the interaction of the nocodazole-tubulin complex with microtubule ends and be the cause of the difference in behavior that we observed in vitro between colchicine and nocodazole.

In vitro assays measuring the interaction of MDAs with tubulin or the functional consequence of this interaction, such as the assembly kinetics, are mainly of mechanistic interest. However, they do not allow to predict the effect of MDAs on cellular microtubules. This effect depends not only on the capacity of these agents to penetrate the cell, but also on the intracellular context. Thus, the action of MDAs can be modulated by efflux pumps, which release therapeutic molecules to the outside, or by a set of regulatory proteins. For example, it has been shown, both in vitro and in cells, that end-binding proteins sensitize MTs to the action of microtubule-targeting agents [1]. The quantitative cell-based assay that we have developed takes into account this cellular context and quantitatively reports the effects of MDAs on cellular MTs. Such a cell-based assay, however, does not allow to know precisely the intracellular concentration of the agent and one cannot exclude the possibility that secondary products may be produced within the cell that could directly act on tubulin/MTs.

In addition, some drugs, such as our recently described compound Carba1, with low affinity for tubulin, may show a visible depolymerizing effect on tubulin assembly kinetics, but no detectable effect on the microtubular network of interphase cells [30]. The cell-based assay described here is thus complementary to in vitro assays on purified tubulin.

This cell-based assay is limited to adherent cell lines and additional steps, such as centrifugations should be implemented for non-adherent cells.

Interpretation of the results of this assay can be more complicated for vinca alkaloids. Indeed, vinca alkaloids have been reported not only to depolymerize microtubules but also to give rise to stable paracrystallin bundles of protofilamentous tubulin, in a time and concentration dependent manner, both in cells [49] and on purified tubulin [50]. Such a reorganization of the microtubule network, generated by high doses of vinca alkaloids (for instance after a 2 h incubation of cells with 25 µM vinblastine), can proceed unnoticed and give a signal equivalent to that of a normal interphase microtubule network [51]. Thus, a simple microscopic analysis of the state of the microtubular network is always useful.

Our results with the reference molecules showed two classes of activity, with C-A4 and vinblastine proving to be more potent in depolymerizing cellular MTs than nocodazole and colchicine, in two different cell types.

There was no significant correlation between the activity of these different drugs on in vitro tubulin assembly and their cellular activity, indicating that the origin of this difference is not the result of a difference in interaction with tubulin. Interestingly, vinblastine and C-A4 have an identical xLogP3—which predicts the compounds‘ lipophilicity- (3.7, source Pubchem, https://pubchem.ncbi.nlm.nih.gov/, accessed on 18 October 2021) whereas xLogP3 of nocodazole and colchicine are lower, with a value of 2.8 and 1 respectively. The different effects of drugs on cellular microtubules therefore most likely reflect their chemical and physical properties in the cellular context.

Finally, although the effect of the drugs on cell viability can result of their interaction with other intracellular targets than tubulin or on metabolized side-products, the striking similarity between the profile of the viability curves and that of the curves measuring the effect of the drugs on the cellular microtubules strongly suggests that the cytotoxic effect observed after 48 h of incubation results from the depolymerizing effect of the drugs on cellular microtubules. However, this conclusion is limited to the compounds analyzed in this study.

## 5. Conclusions

In conclusion, this study shows that the assay we recently developed can bridge the gap between in vitro assays and cell viability assays. Sensitive, easy to implement, this assay does not require a microscope, but a simple microplate reader. It can be miniaturized for high throughput screening and facilitate the discovery and development of novel microtubule poisons and their characterization in the context of living cells. Moreover, it could also find other applications in the context of drug profiling.

## Figures and Tables

**Figure 1 cancers-13-05226-f001:**
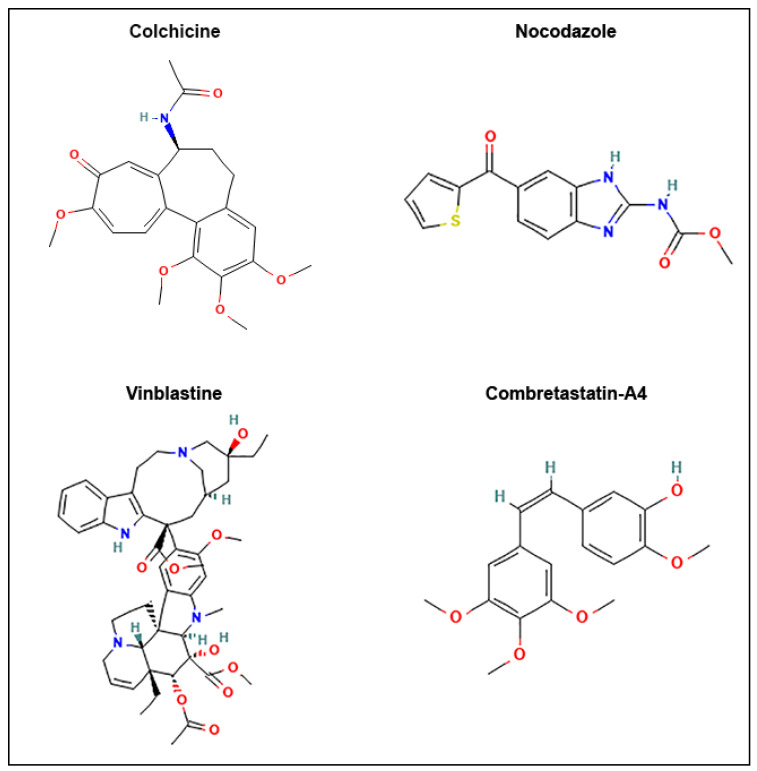
Chemical structure of the microtubule destabilizing agents used in this study.

**Figure 2 cancers-13-05226-f002:**
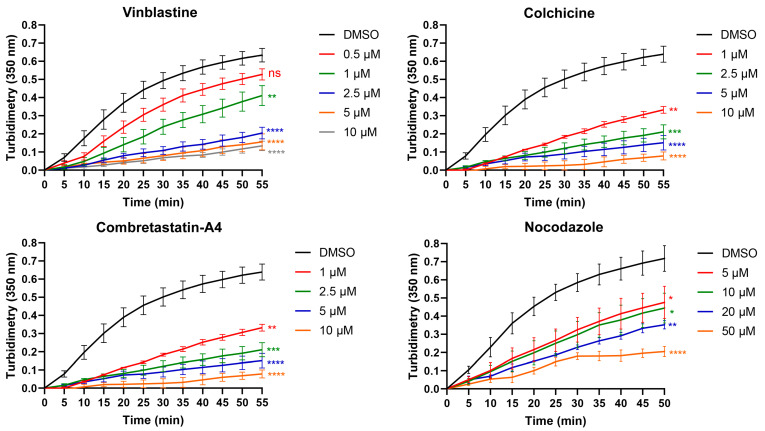
Effect of the reference compounds on the kinetics of in vitro tubulin polymerization. Time course of tubulin polymerization at 37 °C in the presence of vehicle (DMSO, black line) or the different compounds at several concentrations (colored lines) as indicated, measured by turbidimetry at 350 nm. Each turbidimetry value represents the mean ± SEM from 3–4 independent experiments. The significance was determined by a Student’s t-test (* *p* < 0.05, ** *p* < 0.01, *** *p* < 0.001, **** *p* < 0.0001 compared to DMSO).

**Figure 3 cancers-13-05226-f003:**
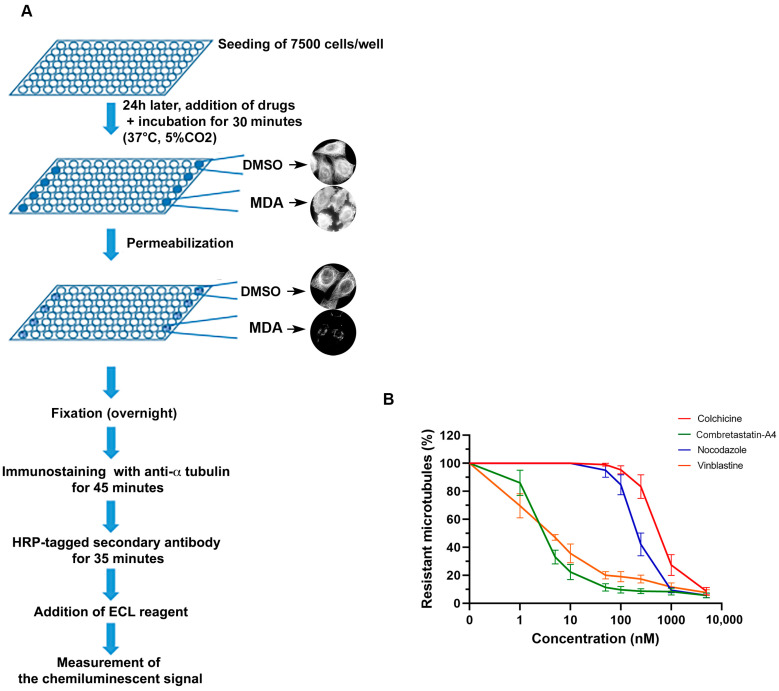
Comparative analysis of the effect of the selected microtubule depolymerizing agents on microtubule contents in HeLa cells. (**A**) Schematic representation of the different steps of the assay. (**B**) Comparison of the MT destabilizing effect of the reference compounds. Different doses of compounds were applied to HeLa cells in microplates and their MT destabilizing effect was assessed after a 30 min incubation, using the luminescent assay, as described in the Materials and Methods section. Results are expressed as % of resistant MTs, with 100% corresponding to cells treated with DMSO only without the depolymerizing agent. Datapoints are means ± SEM from three independent experiments.

**Figure 4 cancers-13-05226-f004:**
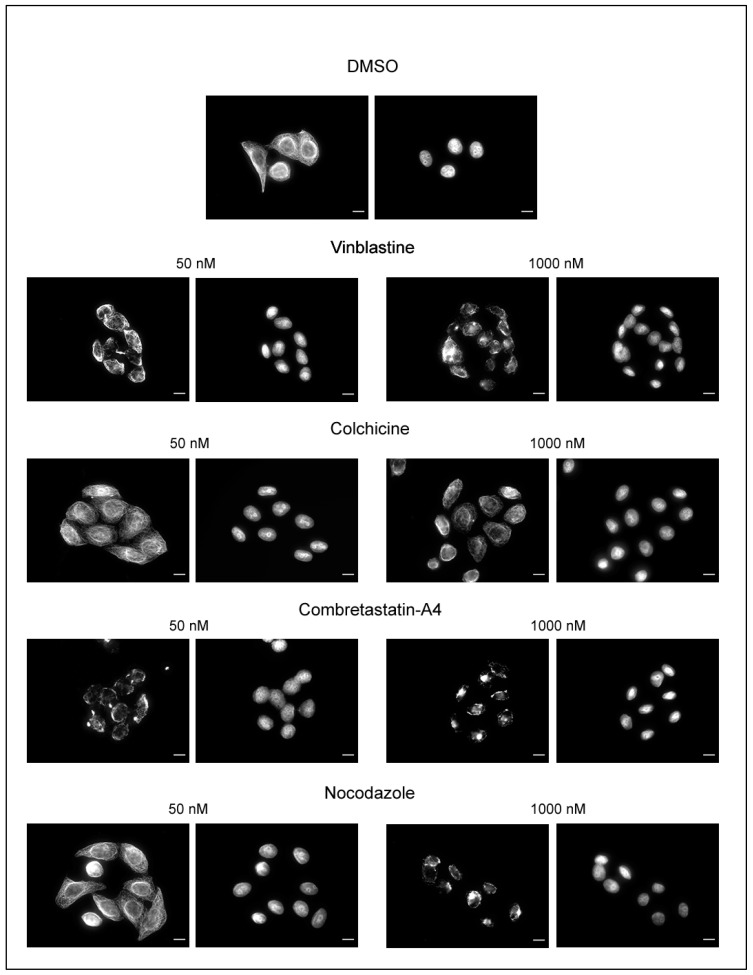
Immunofluorescence analysis of the effect of the compounds on HeLa cells microtubules. HeLa cells were incubated for 30 min with the compounds at the indicated concentrations or with DMSO (control). Cells were then permeabilized and processed for immunofluorescence using an anti-α-tubulin antibody. Nuclei are stained with Hoechst 33258. Scale bars, 10 µm.

**Figure 5 cancers-13-05226-f005:**
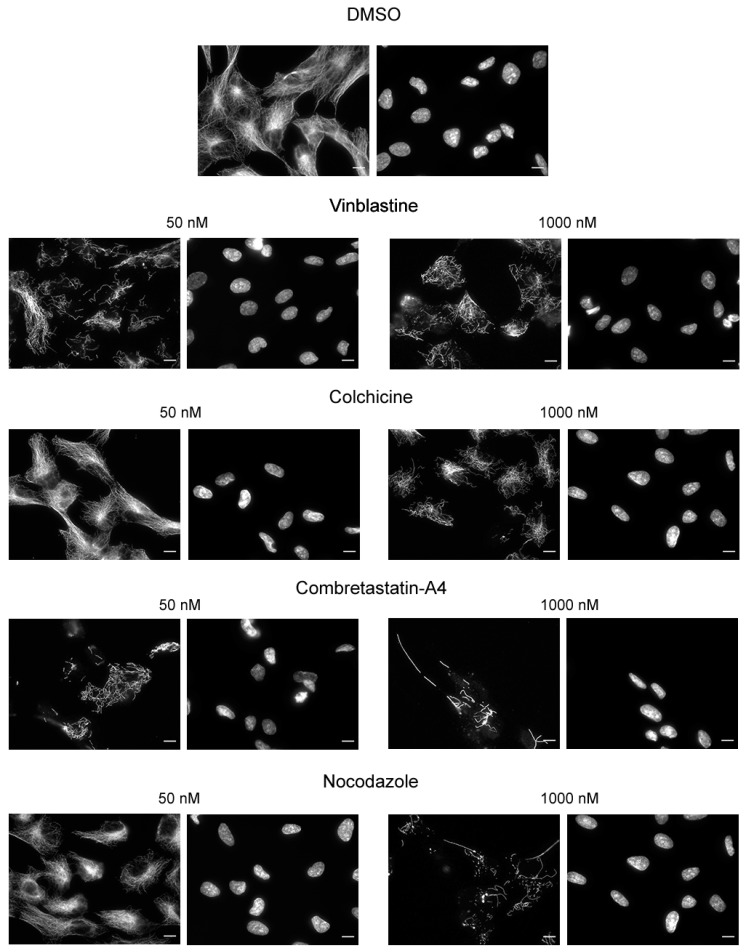
Immunofluorescence analysis of the effect of the reference compounds on RPE1 cell microtubules. RPE1 cells were incubated for 30 min with the compounds at the indicated concentrations or with DMSO (control). Cells were then permeabilized and processed for immunofluorescence using an anti-α-tubulin antibody. Nuclei are stained with Hoechst 33258. Scale bars, 10 µm.

**Figure 6 cancers-13-05226-f006:**
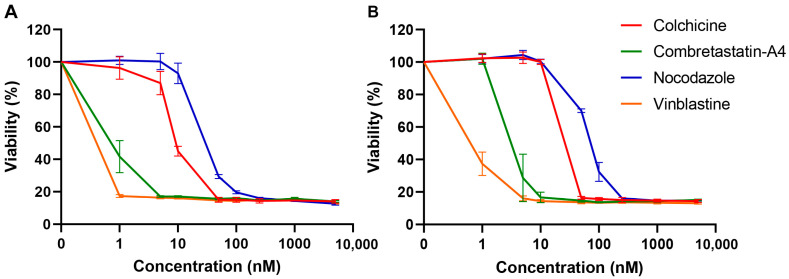
Effect of the compounds on the viability of (**A**) HeLa cells and (**B**) RPE-1 cells. Cells were incubated for 48 h with several concentrations of the indicated compounds. The percentage of viable cells was calculated following a PrestoBlue assay, with 100% corresponding to cells treated with DMSO without the depolymerizing agent. Data are presented as means ± SEM of three independent experiments.

## Data Availability

The data presented in this study are available on request from the corresponding author.

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
