# Peer review of "Characterization of Microtubule Destabilizing Drugs: A Quantitative Cell-Based Assay That Bridges the Gap between Tubulin Based- and Cytotoxicity Assays"

_cancers, 2021, doi:10.3390/cancers13205226_

Round 1

Reviewer 1 Report

The authors Laisne and colleagues report the development of an innovative cell-based assay for better characterize the in vitro activity of microtubule depolymerizing agents. In particular they assessed the activity of four microtubule depolymerizing agents in vitro. Moreover they investigate the effect of the same drugs on cellular microtubules using HeLa cells. Furthermore in order to corroborate the observed data, the authors performed a cell viability assay using presto blue. The authors conclude that the assay can bridge the gap between in vitro assays and cell viability assays and could open the door to a more in depth characterization of novel agents targeting microtubule.

This is an interesting work and could represents a starting point for further investigations.

The following major flaws should be addressed:

  1. In order to assess the activity of microtubule depolymerizing agents a positive control should be added.
  2. Another well-known microtubule interfering drug is represented by eribulin. In this regard, the work titled “Activity of Eribulin in a Primary Culture of Well-Differentiated/Dedifferentiated Adipocytic Sarcoma”. Molecules. 2016 Dec 3;21(12):1662. doi: 10.3390/molecules21121662. PMID: 27918490; PMCID: PMC6273088. should be referenced in the manuscript.
  3. In order to corroborate the observed results MTT, FACS or TUNEL assays should be a good data.
  4. Limitations of the study should be included.

Author Response

Reviewer #1

We thank the reviewer for his constructive comments and we have revised our manuscript accordingly.

All the text modifications are visible using the "Track Changes" function of Word. For clarity, the reviewer’s comments have been copied, pasted and italicized.

The authors Laisne and colleagues report the development of an innovative cell-based assay for better characterize the in vitro activity of microtubule depolymerizing agents. In particular they assessed the activity of four microtubule depolymerizing agents in vitro. Moreover they investigate the effect of the same drugs on cellular microtubules using HeLa cells. Furthermore in order to corroborate the observed data, the authors performed a cell viability assay using presto blue. The authors conclude that the assay can bridge the gap between in vitro assays and cell viability assays and could open the door to a more in depth characterization of novel agents targeting microtubule.

This is an interesting work and could represents a starting point for further investigations.

The following major flaws should be addressed:

  1. In order to assess the activity of microtubule depolymerizing agents a positive control should be added.

We do not understand this point and find it difficult to imagine what positive control the reviewer is thinking of. We used 4 agents whose depolymerizing activity has been known for a long time and widely described. These 4 agents are themselves often used as controls in the literature.

  1. Another well-known microtubule interfering drug is represented by eribulin. In this regard, the work titled “Activity of Eribulin in a Primary Culture of Well-Differentiated/Dedifferentiated Adipocytic Sarcoma”. Molecules. 2016 Dec 3;21(12):1662. doi: 10.3390/molecules21121662. PMID: 27918490; PMCID: PMC6273088. should be referenced in the manuscript.

We have added a paragraph about eribulin (lines 74-77) and cited three additional references (21-23), including the reference proposed by the reviewer.

  1. In order to corroborate the observed results MTT, FACS or TUNEL assays should be a good data.

The aim of our work was not to compare viability tests with each other.  The viability test we used is the prestoblue assay, which, like the MTT, measures the amount of metabolically active cells, but shows lower signal to noise ratio (see for instance Boncler M et al., Comparison of PrestoBlue and MTT assays of cellular viability in the assessment of anti-proliferative effects of plant extracts on human endothelial cells. J Pharmacol Toxicol Methods. 2014;69(1):9-16. doi:10.1016/j.vascn.2013.09.003). Moreover, the effects of the compounds on the cell cycle and apoptosis have already been extensively described and it does not seem relevant for this study to demonstrate them again by FACS or TUNEL analysis.

Limitations of the study should be included.

In addition to the limitations already mentioned (lines 378-394),  following the reviewer's suggestion, we have added two sentences  (lines 386-3872) and line 413 discussing the limitations of the study.

Reviewer 2 Report

In their manuscript (cancers-1374587) Laisne et al. have evaluated the effect of 4 well characterized microtubule destabilizing drugs, i.e. colchicine, nocodazole, vinblastine and combretastatin-A4, on HeLa cervical adenocarcinoma cells. What is important, the Authors determined content of cellular microtubule by their own recently developed method. The manuscript shows some interesting results of in vitro experiments. The study is highly readable and the experiments have been designed properly. However some modifications are necessary to improve the value of the manuscript.

Special comments:

  1. The significance of the obtained results should be statistically assessed.
  2. In the line 92, the Authors wrote that “cell viability assays allow to determine the number of healthy cells in a sample”. Usually, in such studies, the effect of drugs on cancer cells is assessed. However, cancer cells are not a healthy ones. Therefore, the term "healthy cells" should be avoided.
  3. In the chapter “Chemical reagents and cells”, the information that RPE-1 is a human retinal pigment epithelial cells and HeLa cell line is derived from cervical adenocarcinomashould be added.
  4. In the chapter 3.1. “Effect of the compounds on in vitro kinetics of tubulin polymerization” Authors indicated IC50 values of examined drugs. These IC50 values have been estimated from the curves. It is not clear from which curves? How the IC50 values for examined drugs were determined? This information should be added to the Material and methods section.

Author Response

We thank the reviewer for his constructive comments and we have revised our manuscript accordingly. We have addressed all of the reviewer's comments.

All the text modifications are visible using the "Track Changes" function of Word. For clarity, the reviewer’s comments have been copied, pasted and italicized.

In their manuscript (cancers-1374587) Laisne et al. have evaluated the effect of 4 well characterized microtubule destabilizing drugs, i.e. colchicine, nocodazole, vinblastine and combretastatin-A4, on HeLa cervical adenocarcinoma cells. What is important, the Authors determined content of cellular microtubule by their own recently developed method. The manuscript shows some interesting results of in vitro experiments. The study is highly readable and the experiments have been designed properly. However some modifications are necessary to improve the value of the manuscript.

Special comments:

1; The significance of the obtained results should be statistically assessed.

With regard to the results of the assembly tests presented in Figure 2, for each drug tested we compared the values obtained at the plateau with the DMSO control (Student's t test). We have modified the figure and the figure legend accordingly.

Furthermore, for the curves presented in figures 3 and 6, our description in the text had indeed remained very qualitative and we are grateful to the reviewer for highlighting this point.  We have now specified in the text the mean IC50 (lines 262-263) and GI50 (line 328-330) values, ± SEMs, in order to be able to compare quantitatively the effects of the different drugs. We have also explained in the Material and methods section how IC50s (lines 159-161) and GI50s (lines 182-184) were calculated. We hope that we have addressed the reviewer's concern, but we are prepared to conduct additional statistical analyses, which the reviewer may wish to indicate to us.

2. In the line 92, the Authors wrote that “cell viability assays allow to determine the number of healthy cells in a sample”. Usually, in such studies, the effect of drugs on cancer cells is assessed. However, cancer cells are not a healthy ones. Therefore, the term "healthy cells" should be avoided.

We thank the reviewer for this pertinent comment. We have replaced "healthy cells" with "living cells" line 98 and deleted the word "healthy" line 98.

3.  In the chapter “Chemical reagents and cells”, the information that RPE-1 is a human retinal pigment epithelial cells and HeLa cell line is derived from cervical adenocarcinoma should be added.

This is indeed useful information. We have added it, lines 137-138.

4. In the chapter 3.1. “Effect of the compounds on in vitro kinetics of tubulin polymerization” Authors indicated IC50 values of examined drugs. These IC50 values have been estimated from the curves. It is not clear from which curves? How the IC50 values for examined drugs were determined? This information should be added to the Material and methods section.

It is true that we were not very clear on this point. We have now added an explanation in the Material and methods section (lines 159-161) and deleted « estimated from the curves » in the main text (line 236).

Round 2

Reviewer 1 Report

The work has been improved and now could be considered for publication.

Author Response

I don't understand, I have already answerd to that reviewer, who is fine with the response.

May be thi is a mean to upload the new manuscript for the academic editor?